# ERP Correlates of Prospective Memory and Cue Focality in Children

**DOI:** 10.3390/brainsci12050533

**Published:** 2022-04-21

**Authors:** Ana B. Cejudo, Cristina López-Rojas, Carlos J. Gómez-Ariza, María Teresa Bajo

**Affiliations:** 1Department of Experimental Psychology, Research Center for Mind, Brain and Behavior, University of Granada, 18011 Granada, Spain; lopezrojas@ugr.es (C.L.-R.); mbajo@ugr.es (M.T.B.); 2Department of Psychology, University of Jaén, 23071 Jaén, Spain; cjgomez@ujaen.es

**Keywords:** prospective memory, cue focality, Event Related Potentials (ERPs), development, children

## Abstract

Prospective memory (PM) is essential in the everyday activities of children because it involves remembering intentions for the future, such as doing their homework or bringing written parental permissions to school. Developmental studies have shown increases in PM performance throughout childhood, but the specific processes underlying this development are still under debate. In the present study, event-related potentials were used to examine whether the focality of the PM task is related to the PM increments by testing two groups of children (first and last cycle of primary school) and assessing differences in N300 (cue detection), frontal positivity (switching), parietal positivity (retrieval of the intention) and frontal slow waves (monitoring of the retrieved intention). The results showed significant differences in focality in the group of older children but no differences in any of the components for their younger counterparts. In addition, the differences between prospective and ongoing trials were smaller for younger than older children. These findings suggest that the ability to adjust attentional strategies, monitor, switch and retrieve the intention develops across childhood and affects PM performance in attentionally demanding conditions.

## 1. Introduction

Prospective memory (PM), the ability to carry out planned activities in the future [1], is critical for everyday activities during adulthood and childhood (e.g., remembering to bring materials to school). Performance in PM tasks involves both remembering an intention to do something (the prospective component) and retrieving what the intended action is (the retrospective component) [2,3]. PM can be especially difficult during childhood because it requires many demanding executive control processes, which are dependent on efficient functioning of the prefrontal cortex. Thus, executive processes—such as maintaining the intention while performing other ongoing activities, detecting the appropriate moment to perform the intention and stopping the ongoing task (OT) to retrieve and perform that intention—are involved in many PM situations [4]. In fact, some studies suggest a continuous development of the processes underpinning PM across childhood and adolescence that might also be dependent on brain development (see [3,5,6,7,8]). Whereas PM has been widely investigated in young and older adults [9], the relative role of these processes has been much less investigated in young children (see [10], for a theoretical review).

Event-related potentials (ERPs) have been used to dissociate the neurocognitive mechanisms that underlie PM [11,12,13]. For example, both N300 and frontal positivity have been associated with prospective components. N300 is a negative deflection over the occipital and parietal regions that occurs between 300 and 500 ms after the stimulus onset [14,15]. The amplitude of N300 is greater for PM hits than for PM misses and ongoing (ON) trials [14,16], and it has been suggested that N300 reflects the detection of a PM cue [8,14,16]. Interestingly, Mattli et al. [6] reported an age-related pattern in N300, with young adults showing differences between PM hits and PM misses, but with children aged from 10 to 11 not showing these differences. According to the authors, the children’s poorer PM performance might stem from difficulties with cue detection. In contrast, Hering et al. [17] (see also [18]) failed to observe differences in N300 amplitudes between PM hits and ON trials in young adults, although such differences were observed in adolescents. The reasons for the discrepancy between the two studies are not evident, but they might be related to the ages of the participants. While the younger children in Mattli et al.’s [6] study could still be inefficient at detecting PM cues, the adolescents in Hering et al.’s [17] study might have already developed this ability. Hence, the N300 component seems to be able to capture developmental differences regarding cue detection.

Similarly, frontal positivity, a positive deflection over frontal regions occurring between 300 and 500 ms after the stimulus onset, differs between PM trials, ON trials and PM miss trials [6,14,19], and has been linked to retrieval processes related to cue recognition [20] as well as to switching from the OT to the PM task. The latter interpretation is based on a study by Bisiacchi et al. [19], in which modulation of this component was observed only in a task-switch version of the PM task (participants were asked to stop responding to the OT when they detected the PM cue), in contrast to a dual-task version in which participants were asked first to respond to the OT when the PM cue was detected and then to perform the PM intention. As mentioned, in a developmental study, Mattli et al. [6] observed that for both children and adults, PM hits differed from PM misses and OT trials in terms of the frontal positivity component, whereas N300 differences between PM hits and PM misses were present only in adults. These authors suggested that the poorer PM performance of children might stem from difficulties with task switching. While developmental studies have examined age differences in N300 and frontal positivity (e.g., [18]), very few studies have addressed age differences in these ERP prospective components in children younger than 10 years old. Therefore, it is worth examining whether and how these components vary at younger ages (for a review, see [10]).

Age differences in PM trials (relative to ON trials) have also been reported on the basis of two retrospective ERP components, namely, parietal positivity (a sustained positivity over the parietal region that begins at approximately 400 ms when comparing PM and OT conditions) and frontal slow waves (a positive activity over the frontal and parietal regions that begins at approximately 400 ms after the stimulus onset). In PM tasks, these two components have been associated with retrieval and monitoring of the intention, with frontal slow waves varying with the number of intentions and response demands [21,22]. Developmental studies have found age-related differences in the parietal positivity component, with larger differences in PM–OT amplitude in younger participants (adolescents and children from 12 to 13 years old) than in older ones [6,8,17,18]. Studies on frontal slow waves that have compared younger and older adults have found reliable differences between the PM trials in which the intention was remembered and those in which it was forgotten in younger adults only, which suggests difficulties in the retrospective PM component in older people [23,24]. To our knowledge, frontal slow waves have not previously been examined in children.

In sum, although some ERP components have been shown to capture age differences in PM, the results do not always show clear developmental patterns, and very few studies have included young children as participants (for an exception, see [25]). Hence, the main goal of the present study was to contribute to the understanding of PM development throughout childhood. Specifically, our study included two groups of children differing in age. The first group was comprised of children enrolled in the first cycle of primary school in the Spanish educational system (*M*_age_ = 7.8 years, *SD* = 0.32), whereas the second group was comprised of children enrolled in the last cycle of primary school (*M*_age_ = 10.6 years, *SD* = 1.05). We selected these two age groups because developmental research suggests that executive functions gradually develop from childhood through adolescence and adulthood and that it is possible to observe more specialised and efficient executive functioning during the late (relative to early) childhood years [26]. As mentioned, many specialised executive functions, such as goal maintenance, cue monitoring and switching, have been shown to play an important role in PM conditions [27,28,29]. Therefore, examining ages in which these functions are expected to change and specialise is highly relevant.

In addition, in the present experiment, we manipulated the focality of the PM cues [30,31]. The rationale behind this manipulation was to vary the involvement of executive functioning in the PM task so that age differences might be more evident in more demanding situations where the task involves specialised executive processing. Focal PM tasks are those in which the PM cue is part of the OT (i.e., if the OT is a lexical-decision task, the focal cue could be a specific word, such as ‘tortoise’), whereas non-focal PM tasks involve cues that are not part of the OT (i.e., a change in colour of the screen frame while words are being categorised). Current PM theories assume that focal and non-focal tasks differ in their monitoring and/or retrieval demands [32,33], with focal cues involving spontaneous retrieval (without strategic monitoring) and non-focal cues involving strategic monitoring and intentional retrieval. According to the attention to delayed intentions theory [32], cue focality will affect the monitoring processes subserving cue detection and spontaneous PM retrieval. In contrast, the dual-pathway framework [33] assumes that cue focality affects the interplay between monitoring and retrieval, with focal cues involving spontaneous retrieval (without strategic monitoring) and non-focal cues involving strategic monitoring and intentional retrieval. Although theories differ in the locus (monitoring or monitoring/retrieval) of the focality effect, they agree that strategic/attentional processes are differentially involved in focal and non-focal tasks, with lower involvement in the focal than non-focal condition [32,34,35]. Consistent with these assumptions, behavioural experiments have found focality effects (reduced performance for non-focal vs focal tasks) in young adults [36], adolescents [37] and children [31,38]. EEG experiments with young adults have also suggested that non-focal tasks are more difficult and control-demanding than focal tasks. For example, in an EEG study with adult participants, Cona et al. [20] observed larger amplitudes in the frontal positivity component in focal than non-focal conditions but no differences in N300, suggesting automatic recognition of the cue in both conditions but more involvement of the switching processes in the focal than in the non-focal condition. In addition, more negative frontal slow waves in the non-focal than in the focal PM condition suggest more effortful retrospective retrieval and monitoring processes for the non-focal than the focal tasks. Therefore, these processing differences between the types of PM tasks are relevant for exploring age differences in PM [25]. The manipulation of cue focality while recording EEGs in children of different ages might be a suitable strategy to identify the processes underlying the development of PM.

Hence, we asked the two groups of children to categorise images that appeared on the screen as animals or not animals (OT). In the focal condition, children were asked to interrupt the categorisation task whenever images of a ball or kite (prospective cues) appeared on the screen and press the designated key on the keyboard. In the non-focal condition, they were also asked to categorise images as animals/non-animals, but they had to interrupt this ongoing activity and press the assigned key whenever the border of the screen changed to magenta or grey. Since this particular task has been previously used with young children to study focality effects [31], we adapted it for EEG recording.

Similar to previous studies [31,38], we expected age differences in PM performance, especially in the non-focal condition, with older children performing better than younger ones. More importantly, we expected age and focality to modulate the prospective ERP components (N300 and frontal positivity) reflecting cue detection and switching. As mentioned, executive functioning develops throughout childhood, with older children showing more specialised and efficient executive abilities. Therefore, as long as switching and monitoring are critically involved in prospective tasks and modulated by the focality of PM cues, developmental differences should be evident in behavioural and neural patterns as a function of the focality of the task. In addition, we also expected developmental and focality differences in the retrospective components. Parietal positivity has been related to retrieval, whereas frontal slow waves have been related to monitoring and evaluation of the retrieved intention, with these two components varying with focality condition. Hence, as long as retrieval and evaluation of the intention is also subject to development, we expected more difficult maintenance and retrieval of intention processes for younger than older children, particularly in the more demanding non-focal condition. However, we recognise that the exact pattern of effects is difficult to predict because no previous study has involved children as young as 7 years old, and no previous ERP developmental study has directly compared focal and non-focal conditions.

## 2. Materials and Methods

### 2.1. Participants

An a priori analysis (power = 95%, α = 5%, *f* = 0.33) using G*Power [39] revealed that a sample size of 22 participants was sufficient to obtain a statistically significant interaction in a 2 × 2 mixed ANOVA. Children were recruited from a public primary school in Granada (Spain). The younger group included 23 children from the first cycle of primary school (*M_age_* = 7.80, *SD* = 0.32) and 25 from the third cycle of primary school (*M_age_* = 10.65, *SD* = 1.05). Excluded from the analyses were 17 children (9 in the younger group and 8 in the older group) due to poor overall EEG quality. In accordance with the Declaration of Helsinki, caregivers of all the children gave written informed consent to participate in the study after being informed of its general aim. The procedure received approval from the ethics committee of the University of Granada (registration number: 84/CEIH/2015). All participants were Spanish and belonged to families of medium socioeconomic status. The parents of the children were asked to inform us whether their children suffered from any disease or learning impairment conditions. Only children without relevant clinical and medical conditions were included in the study.

### 2.2. Materials

Testing was conducted in one session (lasting approximately 50 min) in an individual room of the Memory and Language Lab at the Research Center for Mind, Brain and Behavior (University of Granada). The session contained two blocks corresponding to the focal and non-focal conditions. The order of the blocks was counterbalanced so that the same number of participants received each condition first. To prevent children from becoming tired during testing, there were three 3-min breaks (one in the middle of each condition and one between conditions).

#### 2.2.1. PM Tasks

The OT consisted of categorising pictures as animals or non-animals. We included 90 images taken from the study by Rossion and Pourtois [40]. Each image was repeated three times in each condition, so that each image was presented six times across conditions. In addition, we used 63 images for OT assessment and practice (for details, see [31]). The stimuli appeared in the centre of the screen surrounded by a 15 by 15 pixel colour border, which randomly changed for each presentation of the stimuli (red, blue, green or yellow) (see Figure 1). The children were asked to press the ‘no’ key on the keyboard (placed on the ‘S’ on the keyboard) if an animal stimulus appeared and the ‘yes’ key (placed on the ‘A’) if a non-animal stimulus appeared. In the prospective focal task, in addition to the OT, the children were asked to remember to press a different key if a target picture (ball or kite) appeared. If a ball appeared, they were to press the ‘square’ key (placed on the ‘L’), and if a ball appeared, they were to press the ‘star’ key (placed on the ‘K’). For the non-focal task, the children were asked to press a different key when the picture frame had a particular colour (magenta or grey). If the screen border changed to magenta, they were asked to press the ‘star’ key (placed on the ‘K’), and if the border was grey, they were asked to press the ‘square’ key (placed on the ‘L’).

Note that the number of intentions to retrieve was kept to a minimum. This is different from most studies with adults, which include more than two intentions, with intentions changing across blocks of trials. This was decided because a pilot study with children suggested that increments in the number of intentions made the task too difficult for younger children.

The experiment had the following structure. The children received instructions regarding the OT and practised it for nine trials, followed by 26 experimental OT trials. Once they finished the single OT block, the children received instructions regarding one of the PM tasks (focal or non-focal), which they practised by performing the OT task that included four PM targets. If they correctly responded to two of the four PM targets, they started the PM task; otherwise, they started another practice cycle. In each PM condition, we used 30 prospective targets inserted into a series of 300 OT trials (see Figure 1). We selected this frequency of PM trials based on previous studies with children of similar ages [41,42]. The PM trials randomly appeared after six, eight or ten OT trials. The PM targets differed for the focal and non-focal conditions. Thus, for the focal condition, half of the PM trials were a kite and the rest were a ball. For the non-focal condition, half of the PM target’s frames were purple and the rest were grey. Half of the children participated in the focal condition first while the other half participated in the non-focal condition first.

The duration of the stimulus presentation for the OT and PM cue trials was set to a minimum time of 1600 ms and a maximum of 2800 ms. When participants responded in 1600 ms or longer, the next trial occurred after an inter-stimulus interval (ISI) presented for 250 ms. When participants responded in less than 1600 ms, they were shown a white screen for 1600 ms and then an ISI. When a participant did not respond within 2800 ms, only the ISI appeared. The task described in this section was carried out on a computer using the E-Prime 2.0 software.

#### 2.2.2. EEG Recording and Pre-Processing

The EEG signal was recorded continuously while the children performed the PM task. The acquisition was conducted using the Synamps2 (Neuroscan, El Paso, TX, USA) and 40 Ag/AgCl electrodes distributed on the scalp at a sampling rate of 1000 Hz. All electrodes were referenced offline to the average of both mastoids, and only electrodes with impedances below 15 kΩ were considered. The data processing was performed with EEGLAB 14.1 [43] running in a Matlab environment (Version 7.4.0; MathWorks, Natick, MA, USA). The EEG data were bandpass filtered between 0.5 and 1000 Hz during online recording. A high-pass filter of 0.1 Hz and a low-pass filter of 30 Hz were also applied to the data offline. We applied a notch filter of 50 Hz to clean the electronic noise in the signal. Artefacts were also removed through visual inspection. Channels with a high level of artefacts were detected by careful visual inspection and interpolated from neighbouring electrodes. The temporal windows were located at the appearance of the stimulus, that is, when the cue appeared. The ERP analysis epoch included 100 ms of pre-stimulus baseline and 1200 ms of post-stimulus activity. Artefact correction was performed using the independent component analysis (ICA) toolbox in EEGLAB for semi-automatic artefact removal. The epoch rejection was performed with a cut-off of ±100 μV (<25% per participant).

ERPs were then averaged by considering four types of trials: (1) Ongoing trials that immediately preceded a focal PM cue (younger children: *M* = 28.48 trials, *SD* = 1.41; older children: *M* = 29.20 trials, *SD* = 1.10); (2) Focal PM hits (younger children: *M* = 27.74 trials, *SD* = 2.59; older children: *M* = 28.76 trials, *SD* = 1.92); (3) Ongoing trials immediately preceding a non-focal PM cue (younger children: *M* = 27.87 trials, *SD* = 3.34; older children: *M* = 29.36 trials, *SD* = 1.20); (4) Non-focal PM hits (younger children: *M* = 15.04 trials, *SD* = 3.67; older children: *M* = 19.00 trials, *SD* = 4.37).

### 2.3. Data Analysis

#### 2.3.1. Behavioural Analysis

Accuracy and response times were analysed independently for the OT and PM trials. The analyses were carried out on the ON and PM trials for each focality condition in each group. Thus, we performed a 2 × 2 mixed factorial ANOVA with focality condition (focal vs. non-focal) and age (younger vs. older children) as independent variables. When appropriate, Bonferroni correction for multiple post-hoc comparisons was applied.

#### 2.3.2. Electrophysiological Data Analysis

To examine age differences when responding to PM cues while performing the OT, we compared the ERPs when the EEG response was time-locked to OT targets (which appeared before each PM cue) for each focality condition (focal vs non-focal). In addition, these OT targets were compared to those time-locked to PM hits. This was carried out to ensure the same number of trials in each condition and to reduce variability due to changes in attention across the experimental session. Thus, for each PM trial, the previous OT trial was considered for comparison. Based on previous studies (e.g., [24]), we selected a time window from 250- to 450-ms over the centro-posterior regions to locate N300, followed by visual inspection of the wave forms. Similarly, we selected a time window from 350- to 450-ms over the anterior region to reflect frontal positivity. In addition, we selected a time window from 500- to 1200-ms over parietal-occipital regions to capture the parietal positivity component as well as the same time window over frontal regions to capture the frontal slow wave component. Before the actual analysis, we performed non-parametric cluster-based permutation analysis as implemented in the Fieldtrip Matlab toolbox software [44] to select the electrodes for each time window maximising the differences between the PM and OT trials. This procedure allowed for the selection of a particular region of interest (electrode clusters) defined in a data-driven manner and not based on the (sometimes inconsistent) ROIs from previous studies or by assumptions regarding the sampling distribution under the null hypothesis. The results of these analyses indicated that electrodes CP4, P3, PZ, P4, O1, OZ and O2 yielded significant differences (*p* < 0.05) for intervals from 250- to 450-ms. For the 350- to 450-ms time window, the cluster included the electrodes F3, FZ, F4, F8, FC3 and FCZ (*p* < 0.05). Finally, for the 500- to 1200-ms time window, clusters included the electrodes F3, FZ, F4, F8, FC3 and FCZ (*p* < 0.05) in the frontal regions and the electrodes P3, PZ, P4, O1, OZ and O2 (*p* < 0.05) in the parietal-occipital regions.

We averaged the mean amplitudes across electrodes and conditions and introduced them into the ANOVAs performed for each component. To examine cue detection, switching and intention retrieval, we looked at differences between PM and OT trials for the time windows for each component (N300, frontal positivity, parietal positivity and frontal slow waves) as a function of cue focality and age. This was performed with mixed ANOVAs with age (6–9 years and 10–12 years), trial type (PM and OT) and condition (focal and non-focal) as factors.

## 3. Results

### 3.1. Behavioural Results

#### 3.1.1. Performance on the OT

Performance was analysed by conducting 2 (focality condition: focal vs. non-focal) by 2 (age: younger children vs. older children) ANOVAs on the proportion of correct responses and mean reaction times, which comprised cue focality as the within-participant factor and age as the between-participant factor. For the sake of simplicity, only statistically significant effects are fully reported for this and other analyses in this article. Figure 2 shows the accuracy and response times for the OT.

*Accuracy*. The analysis showed that the main effect of the focality condition was statistically significant, *F*(1,46) = 46.75, *MSe* = 0.185, *p* < 0.01, η_p_^2^ = 0.50, indicating that accuracy was greater in the focal condition (*M* = 0.94, *SD* = 0.05) than in the non-focal condition (*M* = 0.86, *SD* = 0.08) (all other effects and interactions had *p* > 0.29).

*Response times*. The analysis indicated that the main effects of focality condition, *F*(1,46) = 16.61, *MSe* = 364162, *p* < 0.01, η_p_^2^ = 0.27, and age, *F*(1,46) = 18.06, *MSe* = 758988, *p* < 0.01, η_p_^2^ = 0.28, were reliable, showing faster response times for OT in the focal condition (*M* = 947, *SD* = 221) than in the non-focal condition (*M* = 1072, *SD* = 175) and for older children (*M* = 924, *SD* = 215) compared to younger children (*M* = 1102, *SD* = 126). The interaction was not significant (*p* > 0.223).

#### 3.1.2. Performance on the PM Task

To examine the effect of cue focality on PM performance, we looked at accuracy and reaction time by comparing focal and non-focal trials in each age group. Then, a 2 (focal vs. non-focal) by 2 (age: younger vs. older children) mixed factorial ANOVA was conducted for each measure. Figure 3 shows the mean and standard deviations for each focality by age condition in the PM task.

*Accuracy*. The analysis indicated that the main effect of focality condition, *F*(1,46) = 370.436, *MSe* = 2.930, *p* < 0.01, η_p_^2^ = 0.89, was statistically significant, indicating that the accuracy was greater in the focal condition (*M* = 0.94, *SD* = 0.06) than in the non-focal condition (*M* = 0.59, *SD* = 0.14). The main effect of age did not reach statistical significance (*p* > 0.09), but the interaction focality by age did, *F*(1,46) = 9.393, *MSe* = 0.074, *p* = 0.004, η_p_^2^ = 0.170. As shown in Figure 3, differences between younger and older children were evident in the non-focal condition, *t*(46) = −2.52, *p* < 0.05, *d* = −0.73, but not in the focal condition, *t*(46) < 1, *p* = 0.456, *d* = 0.22 (see Figure 3).

*Response times*. The main effects of focality condition, *F*(1,46) = 87.456, *MSe* = 2672728, *p* < 0.01, η_p_^2^ = 0.66, and age, *F*(1,46) = 53.077, *MSe* = 3330079, *p* < 0.01, η_p_^2^ = 0.54, reached statistical significance, indicating faster response times in the focal condition (*M* = 1162, *SD* = 333) than in the non-focal condition (*M* = 1499, *SD* = 231) and in older children (*M* = 1152, *SD* = 248) compared to younger children (*M* = 1525, *SD* = 163). Interestingly, the interaction condition by age approached statistical significance, *F*(1,46) = 3.195, *MSe* = 97646, *p* = 0.08, η_p_^2^ = 0.06. While the difference between focal and non-focal trials was reliable for both younger children, *t*(46) = −5.987, *p* < 0.01, *d* = −1.66, and older children, *t*(46) = −7.302, *p* < 0.01, *d* = 0.22, the effect of focality was much greater in the older children than in the younger children (see Figure 3).

### 3.2. Electrophysiological Results

Below we discuss the ANOVAS with focality (focal vs. non-focal), type of task trial (ON trials vs PM trials) and age (younger vs older children) for time periods and electrodes corresponding to the prospective (N300 and frontal positivity) and retrospective (parietal positivity and frontal slow waves) components (Figure 4).

#### 3.2.1. N300

This ANOVA was conducted on the EEG data for the cluster of significant electrodes CP4, P3, PZ, P4, O1, OZ and O2 for the 250- to 450-ms time window (see Figure 4 and Figure 5 for the means and standard deviations per condition). There was a main effect of focality, *F*(1,46) = 31.723, *MSe* = 260.132, *p* < 0.01, η_p_^2^ = 0.41, and type of task trial, *F*(1,46) = 56.370, *MSe* = 399.430, *p* < 0.01, η_p_^2^ = 0.30, indicating more negative amplitude in the focal condition (*M* = 0.09, *SD* = 3.36) than in the non-focal condition (*M* = 2.49, *SD* = 2.57) and in PM trials (*M* = −0.18, *SD* = 3.91) than OT trials (*M* = 2.76, *SD* = 2.02). There was no effect of age (*p* > 0.20), but the interaction between focality and age reached statistical significance, *F*(1,46) = 19.812, *MSe* = 162.458, *p* < 0.01, η_p_^2^ = 0.30. As can be seen in Figure 5, the difference between focal and non-focal conditions was reliable for the older children, *t*(24) = −8.097, *p* < 0.01, *d* = −0.08, but not for the younger children, *t*(22) = −0.745, *p* = 0.464, *d* = −0.21. Similarly, the interaction type of task trial by age, *F*(1,46) = 7.766, *MSe* = 55.030, *p* < 0.01, η_p_^2^ = 0.14, indicated that the difference between the OT and PM trials was larger for the older children, *t*(24) = 7.828, *p* < 0.01, *d* = 2.02, than for the younger children, *t*(22) = 3.11, *p* < 0.01, *d* = 0.98. Finally, statistical significance was reached for the interaction of focality by type of task trial, *F*(1,46) = 61.347, *MSe* = 292.864, *p* < 0.01, η_p_^2^ = 0.57, and the interaction of focality by type of task trial by age, *F*(1,46) = 23.91, *MSe* = 114.14, *p* < 0.01, η_p_^2^ = 0.34.

To qualify the latter interaction, we performed analyses for each focality condition shown in Figure 5. In the focal condition, there was a reliable interaction between age and type of task trial, *F*(1,46) = 19, *MSe* = 163.24, *p* < 0.01, η_p_^2^ = 0.3, with comparisons indicating age differences for PM trials, *t*(46) = 4.51, *p* < 0.01, *d* = 1.99. However, for the OT trials, there were no significant differences between the two age groups, *t*(46) = −0.185, *p* = 0.854, *d* = −0.05. In addition, although the differences between PM and OT were significant for the two age groups, they were larger for the older children, *t*(24) = 9.27, *p* < 0.01, *d* = −3.27, than for the younger children, *t*(22) = 3.30, *p* < 0.01, *d* = 1.16. In contrast, the ANOVA for the non-focal condition did not show the interaction of age by type of trial to be reliable (*p* > 0.2).

#### 3.2.2. Frontal Positivity

The ANOVA for the frontal positivity component was conducted on the EEG data from the frontal electrode cluster, F3, FZ, F4, F8, FC3 and FCZ, for the 350- to 450-ms time window (see Figure 4 and Figure 6 for the voltage means and standard deviations per condition). The main effects of focality condition, *F*(1,46) = 45.072, *MSe* = 183.786, *p* < 0.01, η_p_^2^ = 0.50, and type of task trial, *F*(1,46) = 54.333, *MSe* = 169.383, *p* < 0.01, η_p_^2^ = 0.542, were statistically significant, indicating more positive amplitude in the focal condition (*M* = 0.57, *SD* = 2.05) than in the non-focal condition (*M* = −1.41, *SD* = 1.73) and in PM trials (*M* = 0.54, *SD* = 2.28) compared to OT trials (*M* = −1.36, *SD* = 1.50). The main effects of age and its interaction with type of task trial were not reliable (*p* > 0.10 for both). However, the interaction of focality by age reached significance, *F*(1,46) = 9.752, *MSe* = 39.763, *p* < 0.01, η_p_^2^ = 0.18. As can be observed in Figure 6, this interaction indicates slightly greater differences between focal and non-focal trials for the older children, *t*(24) = 6.693, *p* < 0.01, *d* = 2.13, compared to the younger children, *t*(22) = 2.680, *p* < 0.05, *d* = 0.68. More importantly, significance was reached for the focality condition by type of task trial, *F*(1,46) = 48.431, *MSe* = 112.445, *p* < 0.01, η_p_^2^ = 0.51, and the higher order interaction focality condition by type of task trial by age, *F*(1,46) = 9.64, *MSe* = 22.38, *p* < 0.01, η_p_^2^ = 0.17.

To qualify this interaction, we performed analyses for the focal and non-focal conditions. These analyses yielded a significant interaction between type of task trial and age in the focal condition, *F*(1,46) = 8.47, *MSe* = 29.5, *p* < 0.01, η_p_^2^ = 0.16. This indicates that PM trials had more positive amplitudes for the older children than for the younger children, *t*(46) = −2.71, *p* < 0.01, *d* = −0.78 (see Figure 4A whereas age differences were not significant in the OT trials, *t*(46) = 0.819, *p* = 0.417, *d* = 0.24. This interaction was also due to the larger differences between PM and ON for the older children, *t*(24) = −7.58, *p* < 0.01, *d* = −0.44, than for the younger children, *t*(22) = −5.005, *p* < 0.01, *d* = 0.98.

Different from the focal condition, the non-focal condition did not show an interaction between type of trial and age (*p* > 0.05), but there was a main effect of age, *F*(1,46) = 7.60, *MSe* = 28.49, *p* < 0.01, η_p_^2^ = 0.14, showing more positive amplitude in the younger children (*M* = −0.84, *SD* = 1.67) compared to the older children (*M* = −1.26, *SD* = 1.64).

#### 3.2.3. Parietal Positivity

The corresponding ANOVA with focality, type of task trial and age as factors was conducted on the EEG data from the significant electrodes, P3, PZ, P4, O1, OZ and O2, for the 500- to 1200-ms time window (see Figure 7 for means and standard deviations per condition). The main effect of focality was not reliable (*p* > 0.07). However, the main effects of age, *F*(1,46) = 45.072, *MSe* = 183.786, *p* < 0.01, η_p_^2^ = 0.50, and type of task trial, *F*(1,46) = 54.333, *MSe* = 169.383, *p* < 0.01, η_p_^2^ = 0.542, were statistically significant, indicating more positive amplitude for the younger children (*M* = 2.57, *SD* = 2.15) than for the older children (*M* = −2.38, *SD* = 1.99) and in OT trials (*M* = 4.33, *SD* = 1.89) compared to PM trials (*M* = 1.56, *SD* = 2.60).

There was a significant interaction for age by type of task trial, *F*(1,46) = 7.352, *MSe* = 28.828, *p* < 0.01, η_p_^2^ = 0.14. As can be seen in Figure 7, there were larger PM–ON differences for the older children, *t*(24) = 8.43, *p* < 0.01, *d* = 0.89, than for the younger children, *t*(22) = 5.09, *p* < 0.01, *d* = 0.53. Similarly, the interaction of focality by age, *F*(1,46) = 14.962, *MSe* = 67.035, *p* < 0.01, η_p_^2^ = 0.25, indicated that the difference between the focal and non-focal conditions was reliable in the older children, *t*(24) = −4.432, *p* < 0.01, *d* = −0.69, but not in the younger children, *t*(22) = 1.327, *p* = 0.197, *d* = 0.18.

Finally, statistical significance was reached for the interaction of type of task trial by focality, *F*(1,46) = 43.891, *MSe* = 116.029, *p* < 0.01, η_p_^2^ = 0.49, and the interaction of focality by type of task trial by age, *F*(1,46) = 6.25, *MSe* = 16.52, *p* < 0.05, η_p_^2^ = 0.12 (see Figure 7). Follow-up ANOVAS for the focal and non-focal conditions showed that the interaction between type of task trial and age was not statistically significant for the non-focal condition (*p* > 0.5). However, this interaction was significant for the focal condition, *F*(1,46) = 10.52, *MSe* = 44.5, *p* < 0.01, η_p_^2^ = 0.19, indicating that there were differences between the two age groups in PM trials, *t*(46) = 5.07, *p* < 0.01, *d* = 2.33 (see Figure 4B, but not in OT trials, *t*(46) = 1.77, *p* = 0.08, *d* = 0.51. In addition, although the differences between PM and ON were significant for both age groups, they were larger for the older children, *t*(24) = 8.614, *p* < 0.01, *d* = 4.13, than for the younger children, *t*(22) = 5.802, *p* < 0.01, *d* = 1.13.

#### 3.2.4. Frontal Slow Waves

The ANOVA for the EEG data from the frontal electrode cluster, F3, FZ, F4, F8, FC3 and FCZ (see Figure 8 for means and standard deviations per condition), showed significant main effects of age, *F*(1,46) = 4.75, *MSe* = 21.3, *p* < 0.05, η_p_^2^ = 0.09, and type of trial, *F*(1,46) = 77.656, *MSe* = 135.017, *p* < 0.01, η_p_^2^ = 0.63. A more negative amplitude was found in younger children (*M* = −1.93, *SD* = 1.52) compared to older children (*M* = −1.32, *SD* = 1.38) and in PM trials (*M* = −0.76, *SD* = 1.63) compared to OT trials (*M* = −1.36, *SD* = 1.45). However, the main effect of focality condition was not significant (*p* > 0.09).

Although the highest-order interaction focality by type of task trial and age was not reliable (*p* > 0.16), the interaction of focality by type of task trial was reliable, *F*(1,46) = 38.74, *MSe* = 45.91, *p* < 0.01, η_p_^2^ = 0.11. More importantly, however, the interaction of focality by age reached statistical significance, *F*(1,46) = 10.46, *MSe* = 21.3, *p* < 0.01, η_p_^2^ = 0.19. As shown in Figure 8, this interaction indicated significant differences between focal and non-focal trials in the older children, *t*(24) = 4.315, *p* < 0.01, *d* = 0.97, but not in the younger children, *t*(22) = −0.914, *p* = 0.295, *d* = −1.12. The interaction of type of task trial by age was also significant, *F*(1,46) = 5.48, *MSe* = 9.52, *p* < 0.05, η_p_^2^ = 0.11, indicating significant differences between the age groups in the PM trials only, *t*(46) = −3.65, *p* < 0.01, *d* = −1.05, with more positive amplitudes for older children than for younger children (see Figure 8).

## 4. Discussion

In this study, we aimed to examine developmental differences in children’s PM and to determine the source(s) of these possible differences by looking at ERP components that have been linked to prospective processes, namely, cue detection and switching (N300 and frontal positivity) and retrospective processes, such as retrieval and monitoring of intentions (parietal positivity and frontal slow waves). With this purpose, we manipulated the nature of the cue (focal vs non-focal) to vary the attentional demands of the task, and we asked two groups of children (younger and older) to perform a PM task while EEG was recorded. The manipulation of cue focality together with the EEG recording allowed us to examine developmental changes in the processes underlying performance in focal and non-focal cue conditions.

Following the assumptions of both the attention to delayed intentions and the dual pathway frameworks [32,33], and in line with previous studies [31,38], we expected that age differences would be more prominent for non-focal conditions since the PM cue was not part of the OT and more resources were needed to detect it. The behavioural results confirmed the prediction as age differences in PM performance were evident for the non-focal condition, whereas they were not observed for the focal condition. In general, the children’s performance in the PM non-focal condition was worse than in the focal condition in terms of both accuracy and RT. This is in line with previous results from studies with adolescent and adult participants that indicated that focal cues produced better PM performance [37,45,46].

Similarly, the children’s performance of the OT task was better (higher accuracy and faster response times) in the focal condition than in the non-focal condition. Poorer performance of the OT task has usually been interpreted as the cost of maintaining the intention in working memory and monitoring the environment for PM cues [47,48,49,50,51]. Thus, in the present context, the slower RTs and lower accuracy for the OT can be interpreted as an index of the cost of cue monitoring, which was higher for non-focal than for focal cues. This effect was present in both age groups, although there was a tendency for the younger children to show a larger response time cost in the non-focal condition, which suggests that cue monitoring may be more demanding at younger ages. According to the dual process framework [33], the larger cost for non-focal compared to focal cues is due to increased cue monitoring and switching demands under non-focal conditions. In the focal condition, both the ongoing and PM tasks required semantic categorisation (sorting pictures as animals/non-animals (OT) or as a PM cue (kite or ball)), such that no additional processes are necessary to detect the cue. However, in the non-focal condition, additional processing is needed to detect the colour of the frame, such that in addition to semantic processing (of the OT), perceptual, switching and monitoring processes are necessary to detect changes in frame colour [52], making PM performance more complex and demanding. The tendency for younger children to exhibit slower responses during the non-focal task suggests they have more difficulties confronting these demands. In line with this finding, previous work has related age differences in PM performance to the development of executive attention during childhood [10,41,53]. Hence, it is possible that the larger differences observed in the non-focal condition might be due to the use of specific monitoring and attentional allocation strategies that are not sufficiently developed at an early age [26,54,55]. Future research should address the relationship between executive function development and PM performance under non-focal conditions.

Electrophysiological data were helpful for indicating the specific processes underlying the age differences in PM. The analyses of the N300 component (thought to reflect cue detection) showed more negative amplitudes for focal than non-focal conditions and for the PM than ON tasks. Interestingly, age differences were only present for the PM task under focal conditions, with the older children showing more negative amplitudes than the younger children. In contrast, there were no significant age differences in the non-focal condition. This pattern might suggest differences in the ways that younger and older children attempt to detect PM cues. When the PM cue involved the same type of processing as the OT (focal condition), the older children showed more negative responses to the PM cues than to the ON items and more negative responses than younger children to both PM and ON trials. This pattern may reflect easier detection of the PM cue by the older children in the focal condition, which is in line with the developmental trajectories for cue detection with adolescent and adult participants [17].

In addition, older children seemed to adjust their strategies when a different type of processing was required to perform the task, that is, when the pattern for focal and non-focal conditions differed. Both the longer RTs in the OT non-focal condition (relative to the focal condition) and the lack of PM–OT differences in the N300 modulation suggest that in the non-focal condition, the older participants might have adopted a dual-processing strategy in which they processed both semantic information in the words and perceptual (colour) information in the display to monitor for possible PM cues in every trial, even though this strategy may have been costlier in terms of longer times in the OT. Interestingly, younger children seemed to be unable to adjust their strategies to the requirements of the task because they showed similar times in the OT in both the focal and non-focal conditions and smaller modulation of the N300 component between the two conditions. It is also possible that these younger children adopted a dual-task strategy in both types of trials. However, their decreased performance in the non-focal condition indicates that either this strategy did not help them to detect the cue or that they did not use specific strategies to adjust to the requirements of the task. As mentioned, previous research has suggested that attentional control develops throughout childhood/adolescence, and it is possible that the differences observed here between younger and older children reflect this developmental trend [26,54,55].

The frontal positivity component (which is thought to reflect switching) showed a similar developmental pattern. Although differences between the PM and OT trials were present for the focal and non-focal conditions and for older and younger children, these differences were much larger for the older children in the focal condition. In fact, the older and younger children significantly differed in their neural responses to PM trials but not to ON trials. The fact that the older children more readily responded to the focal PM cue and switched from the OT to the PM task than the younger children is in line with the results of previous studies [26,56,57] and suggests that switching is an executive function that develops from early childhood to adolescence. It is also remarkable that PM–OT differences were not found in the non-focal condition for either the younger or older children. In line with our interpretation, the children may have been using a dual-task monitoring strategy in which every trial involved processing the semantic and perceptual features of all the elements of the display. If so, they would have been switching the focus of their attention to different display elements in every trial. This strategy would then result in little differences between the OT and the PM tasks, since cue monitoring and switching were performed similarly in both types of trials. Again, note that although this costlier strategy might also have been used by the younger children under the focal and non-focal conditions (as no changes in N300 and frontal positivity were observed), the differential performance in the two PM conditions suggests that these younger children may not have been able to adapt and use specific strategies, thus, exhibiting worse overall performance in the more difficult (attentionally demanding) tasks.

The overall pattern of results regarding the two PM prospective ERP components is in line with the assumptions of the attention to delayed intentions and the dual pathways frameworks [32,33] and supports the idea that PM conditions modulate the type of processing involved in PM performance. Thus, PM focal tasks involving less attentionally demanding processes recruit different monitoring and detection processes than non-focal tasks. The findings of the present study agree with those of Cona et al. [20], who found greater frontal amplitudes (frontal positivity) for focal than non-focal conditions in adult participants. In addition, the present results suggest that focality-related changes in attentional allocation develop during childhood. Thus, the younger children did not show the focal–non-focal differences that are usually observed in adult participants and that were also present in the older group.

Similarly, parietal positivity and frontal slow waves, which have been associated with retrieval and monitoring of the intention, also showed remarkable developmental changes that suggest different processing requirements for the focal and non-focal PM conditions. In fact, differences between PM and OT trials only appeared (parietal positivity) or were higher (frontal slow waves) in the focal condition in the older group. Thus, for the focal condition, the older children showed larger differences between the PM and OT trials than the younger children. Interestingly, parietal positivity indicated more positive amplitudes for the OT than the PM trials, while frontal slow waves showed the usual pattern of more positive amplitudes for the PM than OT trials. The more positive amplitude for the OT trials might be due to the requirement in those trials to retrieve and discriminate between two possible responses. When the OT trials required discrimination between two responses depending on the nature of the target (animal vs non-animal), the PM cues were associated with only one response. Therefore, it would seem that once the children detected the cue, they had no difficulty remembering the intention. This is consistent with results showing increments in parietal activation in children performing more difficult semantic discriminations [58]. In contrast, post-retrieval monitoring processes (related to frontal slow waves) seemed to be costlier in the less frequent PM trials [20]. Again, the critical pattern is that when the older children more strongly adjusted their retrieval strategies according to the demands of the tasks and showed differences in their parietal positivity and frontal slow waves for the focal and non-focal conditions, the younger children seemed unable to do so, as they showed no differences between the two PM conditions.

To conclude, the present results reveal a developmental change in PM. Although PM performance showed age-related effects only under non-focal conditions, the electrophysiological data showed amplitude differences between the younger (6–9 years) and older (10–12 years) age groups regarding the ERP components associated with cue detection, switching from the ON to the PM task, intention retrieval and the monitoring process of the intention recovery (N300, frontal positivity, parietal positivity and frontal slow waves, respectively). In addition, the focal and non-focal differences in these components were present only in the older children, suggesting that only those children were able to adjust their monitoring and attentional allocation strategies to the demands of the tasks. These findings suggest that, as shown by other developmental studies, the ability to adjust attentional strategies, monitoring, switching and retrieval develop throughout childhood and affect PM performance in attentionally demanding conditions.

## Figures and Tables

**Figure 1 brainsci-12-00533-f001:**
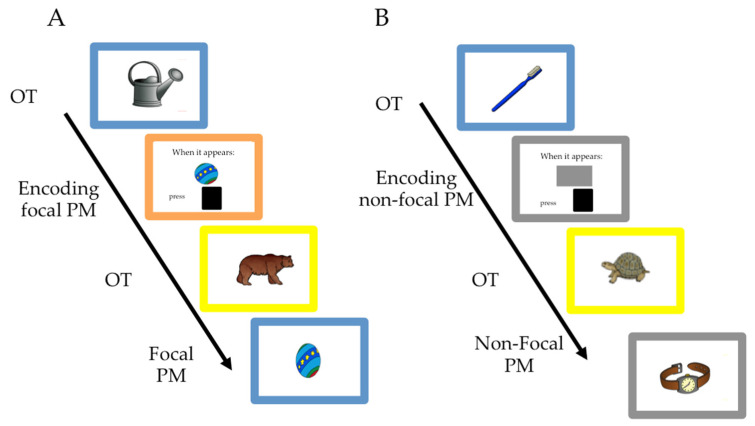
Schematic representation of the experimental paradigm used in the present study. The sequence consisted of practising the ongoing task (OT) and encoding the prospective memory (PM) intention. After the encoding part, participants performed an ongoing block where focal or non-focal PM trials were interweaved. (**A**) Example of a trial sequence for the focal block. (**B**) Example of a trial sequence for the non-focal block.

**Figure 2 brainsci-12-00533-f002:**
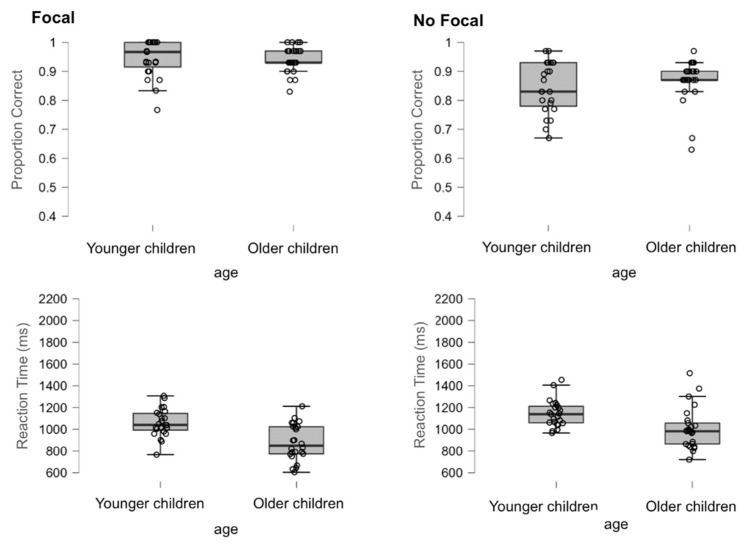
Box plots of correct OT responses (**top** panels) and reaction times (in ms, **bottom** panels) for each group in focal and non-focal conditions.

**Figure 3 brainsci-12-00533-f003:**
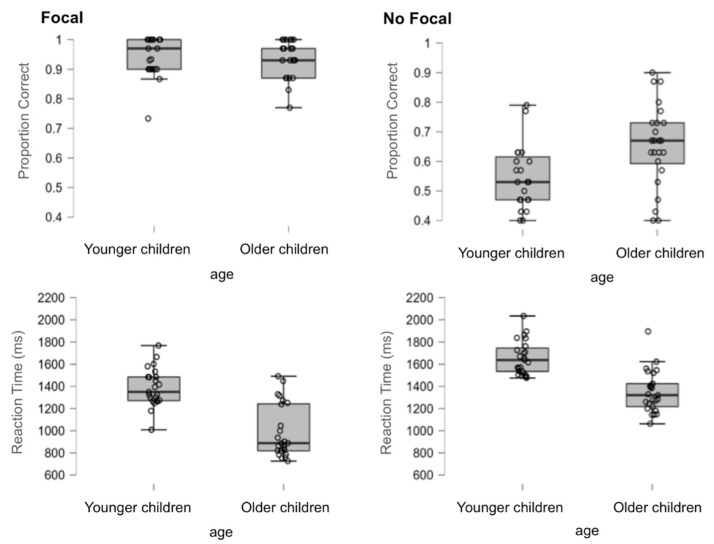
Box plots of correct PM responses (**top** panels) and reaction times (in ms, **bottom** panels) for each group in focal and non-focal conditions.

**Figure 4 brainsci-12-00533-f004:**
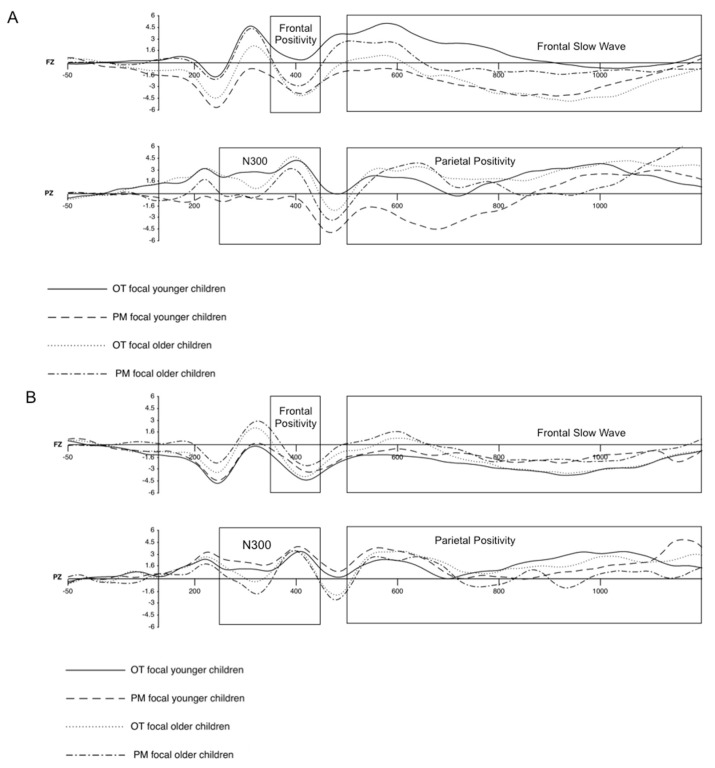
Grand-averaged event-related brain potentials at FZ and PZ electrodes used in the ANOVAs demonstrating frontal positivity, frontal slow waves, N300 and parietal positivity as a function of type of task trial (OT vs. PM) and age group (younger vs older children). (**A**) Grand-average event-related potentials (ERPs) in the focal condition. (**B**) Grand-averaged ERPs in the non-focal condition.

**Figure 5 brainsci-12-00533-f005:**
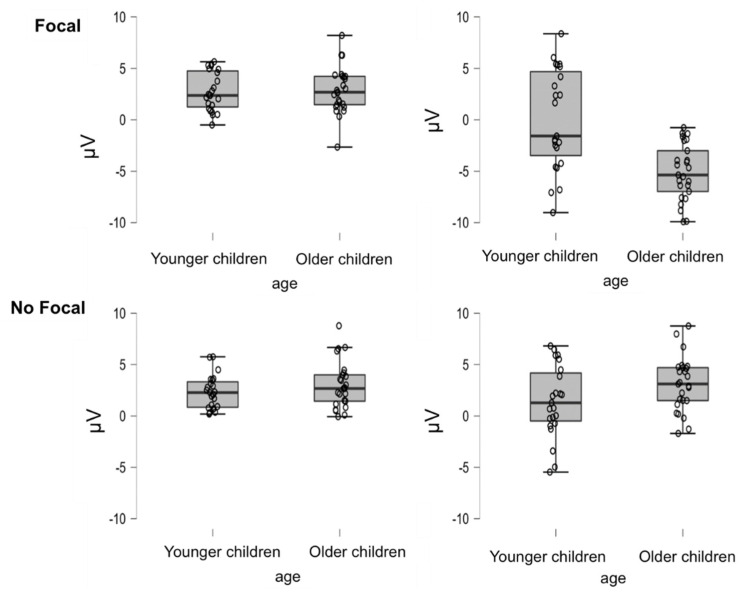
Box plots of the amplitude of the N300 component as a function of age, focality and type of trial.

**Figure 6 brainsci-12-00533-f006:**
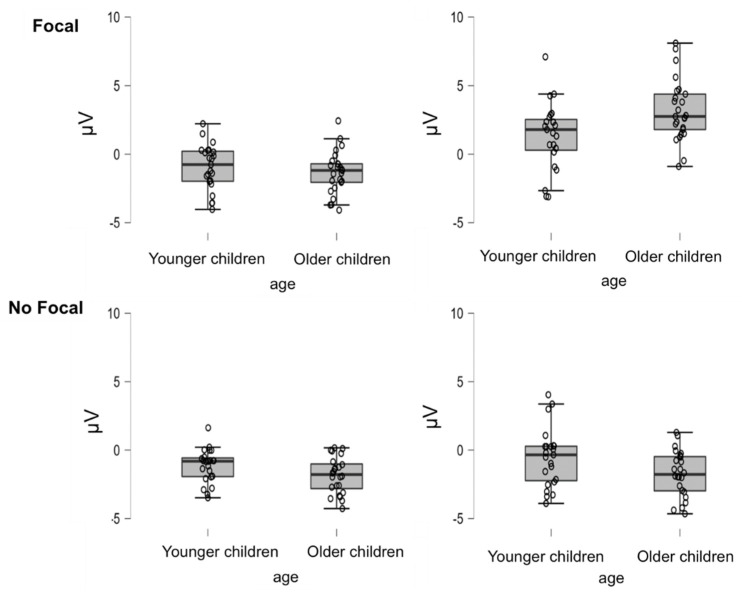
Box plots of the amplitude of the frontal positivity component as a function of age, focality and type of trial.

**Figure 7 brainsci-12-00533-f007:**
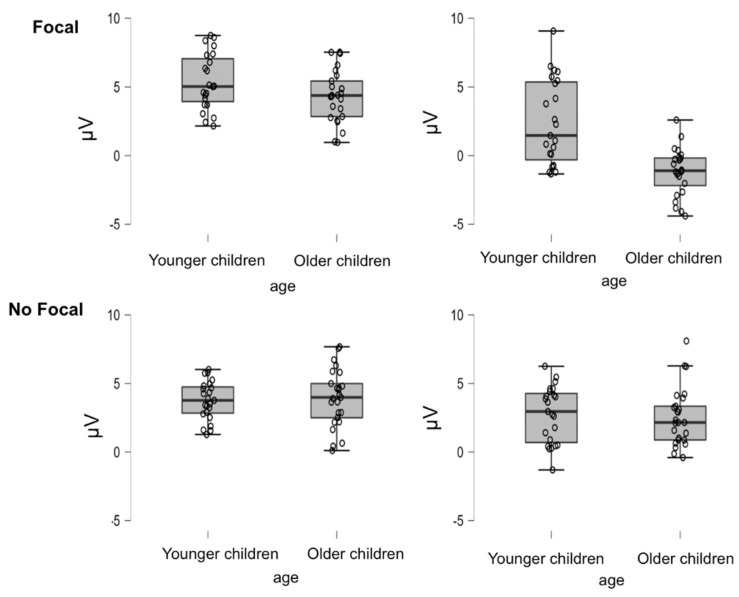
Box plots of the amplitude of the parietal positivity component as a function of age, focality and type of trial.

**Figure 8 brainsci-12-00533-f008:**
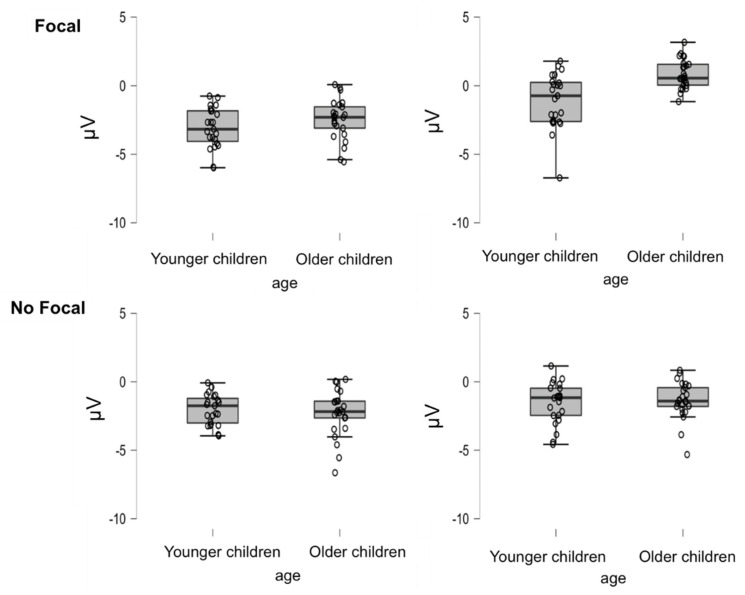
Box plots of the amplitude of the frontal slow waves (FSW) component as a function of age, focality and type of trial.

## Data Availability

Not applicable.

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
