# Peer review of "ERP Correlates of Prospective Memory and Cue Focality in Children"

_brainsci, 2022, doi:10.3390/brainsci12050533_

Round 1

Reviewer 1 Report

This study assesses prospective memory and its different mechanisms in childhood. Authors have found interesting changes in prospective memory between the two ages of children, and the present discovery could deserve publication because of its potential relevance in the field. I have only minor comments.  

Introduction:

We need references after the sentence “Event-related potentials (ERPs) have been used to dissociate the neurocognitive 41 mechanisms that underlie PM.”

We need more accuracy for this sentence: “The reasons for the discrepancies between the two studies are not evident”.

It is difficult to interpret the different hypotheses. I suggest to authors being more accurate on their hypothesis in term of maintenance, retrieval of intention in function of ages. I mean, throughout the manuscript, the reader has to interpret to imagine the different expected findings.

Method:

There is missing words for the following sentence: “We included 90 179 images taken from [37].” From what?

I do not understand why “The children were asked to press the key ‘no’ 184 on the keyboard if an animal stimulus appeared and the key ‘yes’ if a non-animal stim- 185 ulus appeared. “; “no” for what, “yes” for what?

Page 7: “were reliablt » reliable?

During the discussion section, authors introduce the concepts of “the Attention to Delayed Intentions and Dual Pathway frameworks”, but these have never been explained in the introduction section.  

Author Response

I am sending you a revised version of the manuscript number
brainsci-1623682 entitled “ERP correlates of Prospective Memory and Cue Focality in Children ”. We very much appreciate all the constructive comments and suggestions by the reviewer. We detail below the changes that we have introduced to the paper to address them. We have highlighted the changes in the manuscript in red ink. New references have been added to the reference list. We thank the reviewer for the careful reading of our work and the very constructive suggestions.

Thank you for your consideration.

Sincerely,

R1

Introduction:

  1. We need references after the sentence “Event-related potentials (ERPs) have been used to dissociate the neurocognitive 41 mechanisms that underlie PM.”

Thank you for your comment. We have included several references.

Hering, A., Wild-Wall, N., Falkenstein, M., Gajewski, P. D., Zinke, K., Altgassen, M., & Kliegel, M. (2020). Beyond prospective memory retrieval: Encoding and remembering of intentions across the lifespan. International Journal of Psychophysiology147, 44-59.

Wang, Q., Ding, W., Zhang, L., & Chen, G. (2018). The Cognitive and Neural Mechanisms Underlying Event-Based Prospective Memory.

Lamichhane, B., McDaniel, M. A., Waldum, E. R., & Braver, T. S. (2018). Age-related changes in neural mechanisms of prospective memory. Cognitive, Affective, & Behavioral Neuroscience18(5), 982-999.

  1. We need more accuracy for this sentence: “The reasons for the discrepancies between the two studies are not evident”.

Thank you for your comment. Following your suggestion we have expanded our explanation (see pg. 2)

Interestingly, Mattli et al. [6] reported an age-related pattern in N300, with young adults showing differences between PM hits and PM misses, but children ages 10 to 11 not showing these differences. According to the authors, the children’s poorer PM performance might stem from difficulties with cue detection. In contrast, Hering et al. [17; see also 18] failed to observe differences in N300 amplitudes between PM hits and ON trials in young adults, although such differences were observed in adolescents. The reasons for the discrepancy between the two studies are not evident, but they might be related to the ages of the participants. While the younger children in Mattli et al.’s [6] study could still be inefficient at detecting PM cues, the adolescents in Hering et al.’s [17] study might have already developed this ability. Hence, the N300 component seems to be able to capture developmental differences regarding cue detection.

  1. It is difficult to interpret the different hypotheses. I suggest to authors being more accurate on their hypothesis in term of maintenance, retrieval of intention in function of ages. I mean, throughout the manuscript, the reader has to interpret to imagine the different expected findings.

Thank you for your comment. We have reworded a few sentence and included some additional explanations to specify the hypothesis (see pg 4 in red ink)

However, we recognise that the exact pattern of effects is difficult to predict because no previous study has involved children as young as 7 years old, and no previous ERP developmental study has directly compared focal and non-focal conditions.

Method:

  1. There is missing words for the following sentence: “We included 90 179 images taken from [37].” From what?

We have completed the sentence ”… from the work of Rossion and Pourtois”

  1. I do not understand why “The children were asked to press the key ‘no’ 184 on the keyboard if an animal stimulus appeared and the key ‘yes’ if a non-animal stim- 185 ulus appeared. “; “no” for what, “yes” for what?

Thank you for your comment. We apologize if the description of the procedure was not complete or clearly enough. We have now clarified (see pg 4).

The children were asked to press the ‘no’ key on the keyboard (placed on the ‘S’ on the keyboard) if an animal stimulus appeared and the ‘yes’ key (placed on the ‘A’) if a non-animal stimulus appeared. In the prospective focal task, in addition to the OT, the children were asked to remember to press a different key if a target picture (ball or kite) appeared. If a ball appeared, they were to press the ‘square’ key (placed on the ‘L’), and if a ball appeared, they were to press the ‘star’ key (placed on the ‘K’). For the non-focal task, the children were asked to press a different key when the picture frame had a particular colour (magenta or grey). If the screen border changed to magenta, they were asked to press the ‘star’ key (placed on the ‘K’), and if the border was grey, they were asked to press the ‘square’ key (placed on the ‘L’).

  1. Page 7: “were reliablt » reliable?

We have corrected this word

  1. During the discussion section, authors introduce the concepts of “the Attention to Delayed Intentions and Dual Pathway frameworks”, but these have never been explained in the introduction section.  

Thank you for your suggestion, we have completed this part of the introduction (see pg. 3.)

According to the attention to delayed intentions theory [32], cue focality will affect the monitoring processes subserving cue detection and spontaneous PM retrieval. In contrast, the dual-pathway framework [33] assumes that cue focality affects the interplay between monitoring and retrieval, with focal cues involving spontaneous retrieval (without strategic monitoring) and non-focal cues involving strategic monitoring and intentional retrieval.

Reviewer 2 Report

The study is intriguing because it presents a more detailed age comparison to verify the impact of brain maturation on functions that primarily involve executive control, such as prospective memory. However, there are a few things that could be better.

Firstly, I urge a thorough check of the entire text, as there are several grammatical errors and repeated phrases, such as (using) on line 223

The introduction section of the manuscript sufficiently explains all aspects of the study's approach, but there are a few items I'd want to mention in the material and methods sections.

The control of some factors relevant to this analysis, such as the assessment of whether the adolescents had attention deficit hyperactivity disorder or if they were taking any type of medication that could influence response time or reasoning, is not mentioned in the inclusion and exclusion criteria. How did the researchers ensure that these variables were not a confounding factor in the study? The only concern mentioned was the EEG's quality, however a deeper characterization of the sample is required.

The MP task description (sub item 2.2.1) is a little confused, especially when it includes the orientation about the influence of color in the answer option. You might refer to the portion where it is depicted in the diagram and thus indicates the link, in which Figure 1 helps to explain the task. In the written description, where is part A of figure 1? Given the complexity of the task and the large amount of data to be stored, what assurance did you have that the youngster understood everything before beginning the session? I propose that the task description be improved by include the information shown in Figure 1.

Figure 1 – Before explaining the sub-items, I recommend including a title in the legend, as well as the meanings of the acronyms used in the figures.

 The results are very interesting, and the analyses were appropriate, but the way they were presented in a simple table with mean and standard deviation did not allow for a clear analysis of the comparison between groups and conditions. I recommend including or switching to boxplot graphics with jitter and including statistical analysis to help visualize the results more clearly.

Please include the number of participants (N) in the analyses displayed in Tables 3–6, as well as the statistical results. The caption for Figure 2 became long, repeated, and unclear. Please use the letters A and B instead of Top and B, as shown in the illustration, and include the title of the figure before the explanation of its elements.

I believe it would be fair to publish a table with all the raw data as supplemental material with reference work for future study.

Author Response

I am sending you a revised version of the manuscript number
brainsci-1623682 entitled “ERP correlates of Prospective Memory and Cue Focality in Children ”. We very much appreciate all the constructive comments and suggestions by the reviewer. We detail below the changes that we have introduced to the paper to address them. We have highlighted the changes in the manuscript in red ink. New references have been added to the reference list. We thank the reviewer for the careful reading of our work and the very constructive suggestions.

Thank you for your consideration.

Sincerely,

R2

The study is intriguing because it presents a more detailed age comparison to verify the impact of brain maturation on functions that primarily involve executive control, such as prospective memory. However, there are a few things that could be better.

  1. Firstly, I urge a thorough check of the entire text, as there are several grammatical errors and repeated phrases, such as (using) on line 223

The entire text had been revised by a professional translation company.

The introduction section of the manuscript sufficiently explains all aspects of the study's approach, but there are a few items I'd want to mention in the material and methods sections.

  1. The control of some factors relevant to this analysis, such as the assessment of whether the adolescents had attention deficit hyperactivity disorder or if they were taking any type of medication that could influence response time or reasoning, is not mentioned in the inclusion and exclusion criteria. How did the researchers ensure that these variables were not a confounding factor in the study? The only concern mentioned was the EEG's quality, however a deeper characterization of the sample is required.

Thank you for your suggestion. We have completed this information (see pg. 4)

Children’s parents were asked to inform us whether their children suffered from any disease or learning impairment conditions. Only children without relevant clinical and medical conditions were included in the study.

  1. The MP task description (sub item 2.2.1) is a little confused, especially when it includes the orientation about the influence of color in the answer option. You might refer to the portion where it is depicted in the diagram and thus indicates the link, in which Figure 1 helps to explain the task. In the written description, where is part A of figure 1? Given the complexity of the task and the large amount of data to be stored, what assurance did you have that the youngster understood everything before beginning the session? I propose that the task description be improved by include the information shown in Figure 1.

Thank you for your comment. We apologize if the description of the procedure was not complete or clearly enough. We have now clarified it (see pg. 4,).

The children were asked to press the ‘no’ key on the keyboard (placed on the ‘S’ on the keyboard) if an animal stimulus appeared and the ‘yes’ key (placed on the ‘A’) if a non-animal stimulus appeared. In the prospective focal task, in addition to the OT, the children were asked to remember to press a different key if a target picture (ball or kite) appeared. If a ball appeared, they were to press the ‘square’ key (placed on the ‘L’), and if a ball appeared, they were to press the ‘star’ key (placed on the ‘K’). For the non-focal task, the children were asked to press a different key when the picture frame had a particular colour (magenta or grey). If the screen border changed to magenta, they were asked to press the ‘star’ key (placed on the ‘K’), and if the border was grey, they were asked to press the ‘square’ key (placed on the ‘L’).

  1. Figure 1 – Before explaining the sub-items, I recommend including a title in the legend, as well as the meanings of the acronyms used in the figures.

Thank you for the suggestion, we have completed the legend.

Figure 1. Schematic representation of the experimental paradigm used in the present study. The sequence consisted of practicing the ongoing task (OT) and encoding the prospective memory (PM) intention. After the encoding part, participants performed an ongoing block where focal or non-focal PM trials were interweaved. (A) Example of a trial sequence for the focal block. (B) Example of a trial sequence for the non-focal block.

  1. The results are very interesting, and the analyses were appropriate, but the way they were presented in a simple table with mean and standard deviation did not allow for a clear analysis of the comparison between groups and conditions. I recommend including or switching to boxplot graphics with jitter and including statistical analysis to help visualize the results more clearly.

Figure 2. Mean percentage of correct OT responses (top panels) and reaction times (in ms, bottom panels) for each group in focal and non-focal conditions.

Figure 3. Mean percentage of correct PM responses (top panels) and reaction times (in ms, bottom panels) for each group in focal and non-focal conditions.

  1. Please include the number of participants (N) in the analyses displayed in Tables 3–6, as well as the statistical results.

I agree with your suggestion, but this information is included 2.2.2. EEG Recording and Pre-processing section:

“ERPs were then averaged by considering four types of trials: 1) Ongoing trials that immediately preceded a focal PM cue (younger children: M = 28.48 trials, SD = 1.41; older children: M = 29.20 trials, SD = 1.10); 2) Focal PM hits (younger children: M = 27.74 trials, SD = 2.59; older children: M = 28.76 trials, SD = 1.92); 3) Ongoing trials immediately preceding a non-focal PM cue (younger children: M = 27.87 trials, SD = 3.34; older children: M = 29.36 trials, SD = 1.20); and 4) Non-focal PM hits (younger children: M = 15.04 trials, SD = 3.67; older children: M = 19.00 trials, SD = 4.37)”

The caption for Figure 2 became long, repeated, and unclear. Please use the letters A and B instead of Top and B, as shown in the illustration, and include the title of the figure before the explanation of its elements.

Thanks for pointing it out, we have now reworded it (see pg.10)

Figure 2 ...now Figure 4. Grand-averaged event-related brain potentials at FZ and PZ electrodes used in the ANOVAs demonstrating frontal positivity, frontal slow waves, N300 and parietal positivity as a function of type of task trial (OT vs PM) and age group (younger vs older children). (A) Grand-average event-related potentials (ERPs) in the focal condition. (B) Grand-averaged ERPs in the non-focal condition.

  1. I believe it would be fair to publish a table with all the raw data as supplemental material with reference work for future study.

Thank you for your suggestion. Data Availability Statement: All relevant data are available from the Open Science Framework (DOI 10.17605/OSF.IO/8Y74D).

Round 2

Reviewer 2 Report

The article is well-written and covers a topic that is of interest to readers. Despite a flaw in the study design regarding the subjectivity of the children's inclusion criteria, I believe the methodology is adequate, even though no further evaluation was carried out regarding the learning deficit in the children included, which could be a confounding factor in the data interpretation.
The Figures in the results section of the manuscript were much better with the usage of boxplot, but the statistical analysis indicator could have been added as well so that the figure had a complete graphic depiction of the result. Similarly, the findings of items 3.2.2, 3.2.3, and 3.2.4 might have been changed for the boxplot. The discussion and conclusion sections are adequate.

Author Response

As the reviewer suggested,  we have included plots for EEG data, so that descriptive statistics for the different components are now displayed in the Figures 5,6,7,8. All other statitics regarding EEG components are in the text (pgs10-14).
